# The neonatal southern white rhinoceros ovary contains oogonia in germ cell nests

Ruth Appeltant [1,7], Robert Hermes [2], Susanne Holtze [2], Silvia Clotilde Modina[3], Cesare Galli [4,5], Briet. D. Bjarkadottir[1], Babatomisin V. Adeniran[1], Xi Wei[1], Aleona Swegen[1,8], Thomas Bernd Hildebrandt[3,6] & Suzannah A. Williams [1✉]

The northern white rhinoceros is functionally extinct with only two females left. Establishing methods to culture ovarian tissues, follicles, and oocytes to generate eggs will support conservation efforts using in vitro embryo production. To the best of our knowledge, this is the first description of the structure and molecular signature of any rhinoceros, more specifically, we describe the neonatal and adult southern white rhinoceros (*Ceratotherium simum simum*) ovary; the closest relation of the northern white rhinoceros. Interestingly, all ovaries contain follicles despite advanced age. Analysis of the neonate reveals a population of cells molecularly characterised as mitotically active, pluripotent with germ cell properties. These results indicate that unusually, the neonatal ovary still contains oogonia in germ cell nests at birth, providing an opportunity for fertility preservation. Therefore, utilising ovaries from stillborn and adult rhinoceros can provide cells for advanced assisted reproductive technologies and investigating the neonatal ovaries of other endangered species is crucial for conservation.

[1] Nuffield Department of Women's and Reproductive Health, University of Oxford, Women's Centre, Level 3, John Radcliffe Hospital, Oxford, UK. [2] Leibniz Institute for Zoo and Wildlife Research, Alfred-Kowalke-Str 17, D-10315 Berlin, Germany. [3] Dipartimento di Medicina Veterinaria, Università degli Studi di Milano, Via dell'Università 6, 26900 Lodi, Italy. [4] Avantea srl, Laboratory of Reproductive Technologies, Via Porcellasco 7/F, 26100 Cremona, Italy. [5] Fondazione Avantea, 26100 Cremona, Italy. [6] Freie Universität Berlin, D-14195 Berlin, Germany. [7] Present address: Gamete Research Centre, Veterinary Physiology and Biochemistry, Department of Veterinary Sciences, University of Antwerp, Wilrijk, Belgium. [8] Present address: Priority Research Centre for Reproductive Science, University of Newcastle, Callaghan 2308 NSW, Australia. ✉email: suzannah.williams@wrh.ox.ac.uk

As part of the current mass extinction, the northern white rhinoceros subspecies can be considered functionally extinct since just two females remain[1]. Therefore, there is very little time left for scientists to save this subspecies of the white rhinoceros, *Ceratotherium simum ssp. cottoni*, and the use of established advanced assisted reproductive techniques (aART) is necessary[2] along with the development of new technologies. Before being able to apply innovative and challenging aART procedures, knowledge on the basic physiology of the rhinoceros species is crucial. Therefore, this study focused on providing a comprehensive histological characterisation and assessment of adult and neonatal southern white rhinoceros ovaries and their follicular development, which will lead the way to developing in vitro techniques to culture or create rhinoceros' oocytes.

At birth, the ovary contains a reserve of thousands of immature oocytes in primordial follicles[3], a pool that is continually depleted as a female ages. Establishing methods to culture primordial follicles in the laboratory and generate mature eggs would vastly increase the chance of successful in vitro embryo production in endangered rhinoceros species. Assisted reproductive technologies can be beneficial to overcome fertility problems or increase genetic diversity[4,5]. Specifically, captive white, black and Indian rhinoceros often exhibit abnormal reproductive oestrus cycles[6,7] or long non-reproductive periods as opposed to the regular reproductive cycles and fertility observed in the wild[4,8]. When a species is already extinct in the wild, like in the case of the northern white rhinoceros, infertility in ex-situ populations (kept outside of their natural habitat) might have tremendous consequences. Development of aART, such as in vitro germ cell culture, might become crucial to the species survival[9]. Attempts to mature and fertilise rhinoceros' oocytes have been made in the Sumatran (*Dicerorhinus Sumatrensis*) and black (*Diceros bicornis*) rhinoceros and although the low success rate, it showed gamete rescue has promise[10,11].

Reproduction-wise, the aging process, with the incessant depletion of the primordial follicle pool and the concomitant decrease in oocyte quality, does not support good fertility in old age. Some species escape these issues with unusual strategies. For example, the ovaries of the naked mole rat (*Heterocephalus glaber*) contain germ cell nests consisting of the precursors of oocytes, not only at birth but also during a significant portion of adult life[12]. The process of postnatal neo-oogenesis in naked mole rats is the ultimate escape route from a depleting finite ovarian reserve and avoids the reproductive senescence observed in most other species[12]. The continued presence of germ cells after birth in structures known as germ cells nests provides the ideal source material for in vitro gametogenesis. However, as germ cells are rarely available, investigations are ongoing into the potential usefulness of other tissues or cells; e.g., foetal[13] or adult (reviewed by Telfer and Anderson[14]) organs, or embryonic or induced pluripotent stem cells[15,16]. Such research is not only generating crucial knowledge on the production of competent gametes, but is also facilitating the development of innovative reproductive therapies with clinical promise[17] for both humans and endangered species.

As demonstrated by the novel strategies of individual species, a prerequisite for developing in vitro culture methods is a clear understanding of follicular development of that specific species. For various rhinoceros species, although ultrasound of live animals combined with post-mortem histology have been carried out to examine reproductive tract pathologies[4,18–20], basic histological descriptions of the physiology of the healthy rhinoceros ovary are lacking. Crucially, in 2001, a report on the histology of the rhinoceros 'reproductive tract' was published[21], however, it did not include any ovarian histology. With scant knowledge about the structure and function of the rhinoceros ovary, detailed studies are crucial. To address this gap in the knowledge, this study investigated the structural and molecular characteristics of neonatal and adult southern white rhinoceros ovaries. Based on our information, this is the first report describing follicle development in any rhinoceros species.

## Results

**Collection of ovarian tissue**. The adult rhinoceros ovaries were unusually flat unlike those of other species, which are generally spherical or oval in shape. In contrast to most other species, the cortical and medullar part were not oriented as an outer and inner part of the ovary, but rather two-sided with a smooth cortical side and an 'open' and more vascular medullar side (Fig. 1a.i and 1a.iii compared to 1a.ii and 1a.iv, respectively).

**Ovarian structure revealed by histological staining and bright-field scanning**. The one-sided cortex was confirmed histologically (Fig. 2a). Masson's trichrome staining clearly demonstrated a smooth demarcated, cellular cortical side (Fig. 2a.i) that transitioned (Fig. 2a.ii) into the stromal region and collagen containing medullar side (Fig. 2a.iii) (rhino 3). Scanned PAS (Fig. 2b) and HABP (Fig. 2c) sections from the neonate provide an overview of the structure of the ovary. Although there is also an asymmetrical orientation visible, the clear demarcation of cortex and medulla cannot be seen (Fig. 2b, c). Extracellular matrix (ECM) and stromal cells separate multiple areas (Fig. 2c), and these areas, which consist of specific cell populations, comprise the main part of the sample. For each scanned section, a developing follicle is illustrated at higher magnification (Fig. 2b.i and Fig. 2c.i). The polarised structure of the adult ovary is once again evident in scanned whole ovary sections stained with H&E (Fig. 3a), Masson's trichrome (Fig. 3b) and hyaluronic acid binding protein (HABP) (Fig. 3c). Higher magnifications of antral follicles (Fig. 3a.i and Fig. 3c.i) and a primary follicle (Fig. 3b.i) can be recognised.

Follicles were found in all four rhinoceroses. The samples from older animals clearly contained fewer follicles than the neonate (evidenced by the number of sections that needed to be analysed to locate follicles). These results correspond to findings in the Sumatran and black rhinoceros[10,11]. All stages of follicular development from primordial to antral were observed. In the adult ovary, we identified follicles containing an oocyte at every developmental stage except for preantral (Fig. 3d–g). All the normal components of an antral follicle were clearly visible such as oocyte, nucleus, cumulus oophorus, granulosa cells, follicular fluid, basal membrane, theca interna and theca externa (Fig. 3g). In the neonatal ovary, all stages of follicle development are present at birth (Fig. 4). Quantitative characterisation of the different aspects of developing follicles in both neonatal and adult ovaries provided novel insight into follicle development in the southern white rhinoceros (Table 1). The number of granulosa cells ranged from six in an adult primordial follicle to 1955 in an adult antral follicle with a diameter of 786.14 μm (Table 1). The mean oocyte diameter increased from $28.70 \pm 4.03$ μm in secondary adult follicles to 93.08 μm in an adult antral follicle. Since the measured antral follicles are relatively small compared to a pre-ovulatory follicle, we expect the size will increase with further development. It is clear the ratio of oocyte to follicle size decreased during the transition to more developed follicles as occurs in other species. Some follicles in the neonatal ovary had an unusual elliptical appearance where the granulosa cell layers surrounding the oocyte ranged from 1 layer to multiple layers within the same follicle (Fig. 5a.i–iv).

Often follicles were found in 'clusters' rather than homogeneously dispersed throughout the whole ovary (Fig. 5b). This

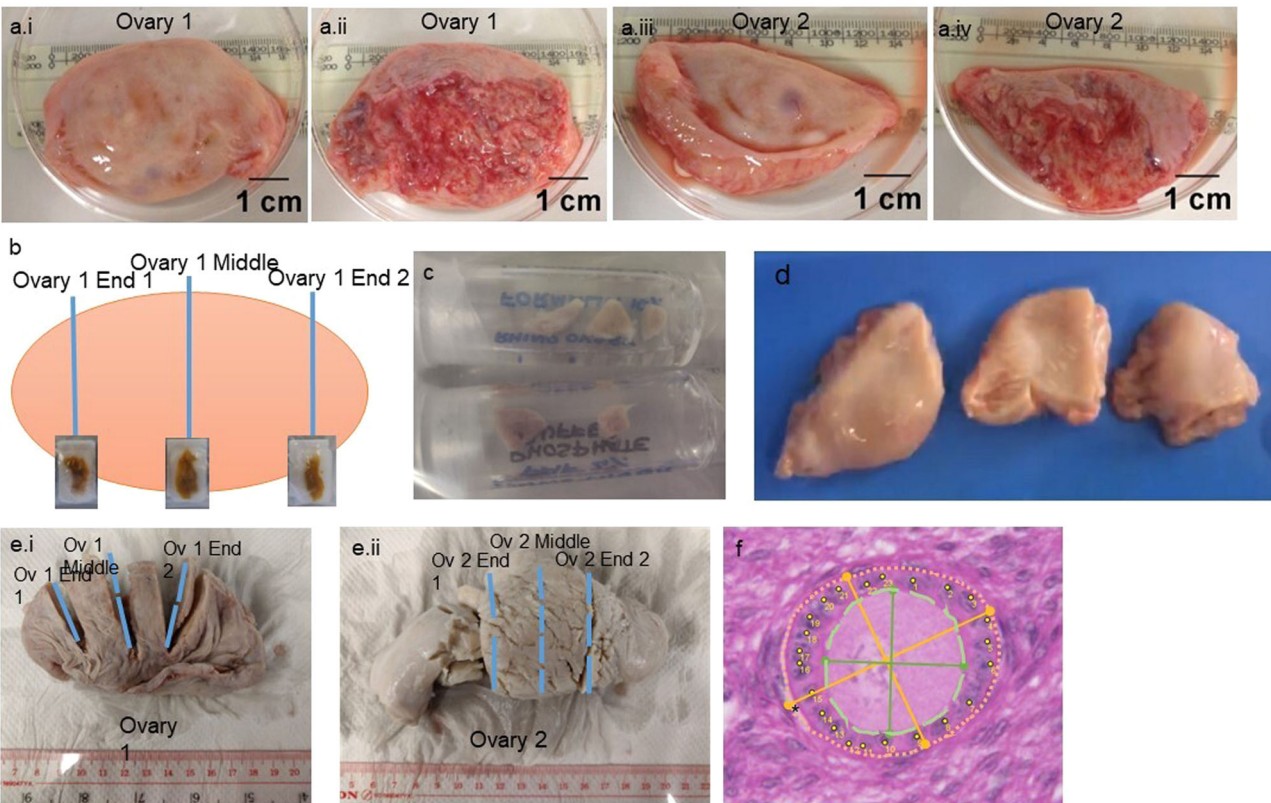

**Fig. 1 The macroscopic view of ovaries of neonatal and adult southern white rhinoceros. a** Adult rhinoceros of 39 years old. **a**.i and **a**.iii are the smooth side of the ovaries, a.ii and a.iv are the opposite medullar side of the same ovaries. **b** Illustration of how cross sections were collected from the ovaries of rhino 2 and 4: a cross section at one end of the ovary, the middle part and the other end. **c** Fixed ovarian samples provided from a neonatal southern white rhinoceros (rhino 1). **d** Three pieces of ovarian tissue obtained from a southern white rhinoceros (rhino 3, age 30 years); the ovaries were in pieces on arrival. **e**.i-ii Ovary 1 and ovary 2 of a southern white rhinoceros (rhino 4, age 38 years). The blue lines indicate how the cross sections were obtained. The broken blue lines indicate where the tissue was separated for embedding. **f** Illustration of the measurements performed on a follicle: longest oocyte diameter and perpendicular diameter (green full line), longest follicle diameter and perpendicular diameter (orange full line), oocyte area (green dashed line), follicle area (orange dotted line), granulosa cell count (yellow labelled dots), shrinkage space (asterisk).

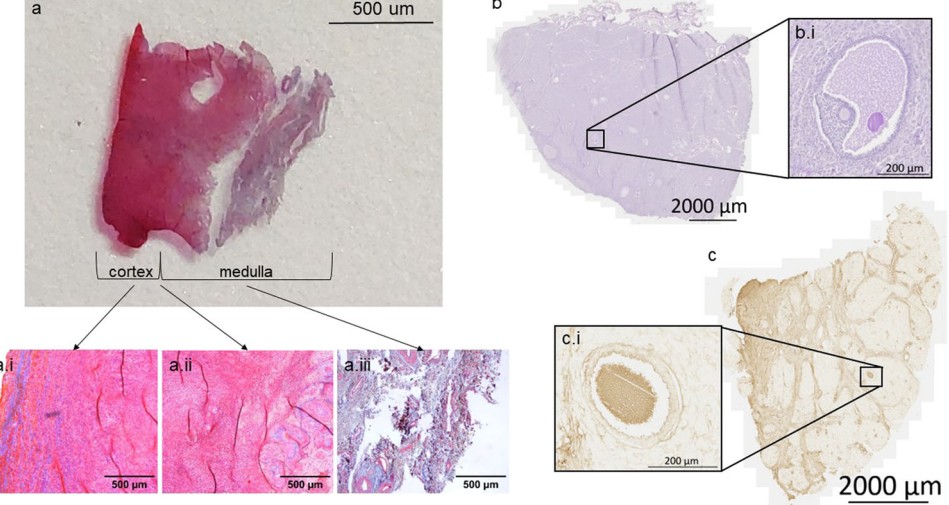

**Fig. 2 Overview of the structure of a neonatal and adult ovary cross section of southern white rhinoceroses. a** Ovarian section from an adult southern white rhinoceros ovary (rhino 3, age 30 years) stained with Masson's trichrome. The intense red area on the left of the tissue is the cortex and the blueish area on the right is the medulla. Images (**a**.i, **a**.ii and **a**.iii) are details of the cortex, the transition between cortex and medulla and the medulla respectively. **b** and (**c**) Sections of a neonatal southern white rhinoceros ovary. The higher magnification images illustrate antral follicles (**b**.i and **c**.i). **b** is stained with Periodic acid–Schiff staining (PAS), (**c**) is detection of hyaluronic acid (HA) using HA binding protein (HABP). In (**c**), the stromal bifurcations dividing the ovary into segments are clearly visible.

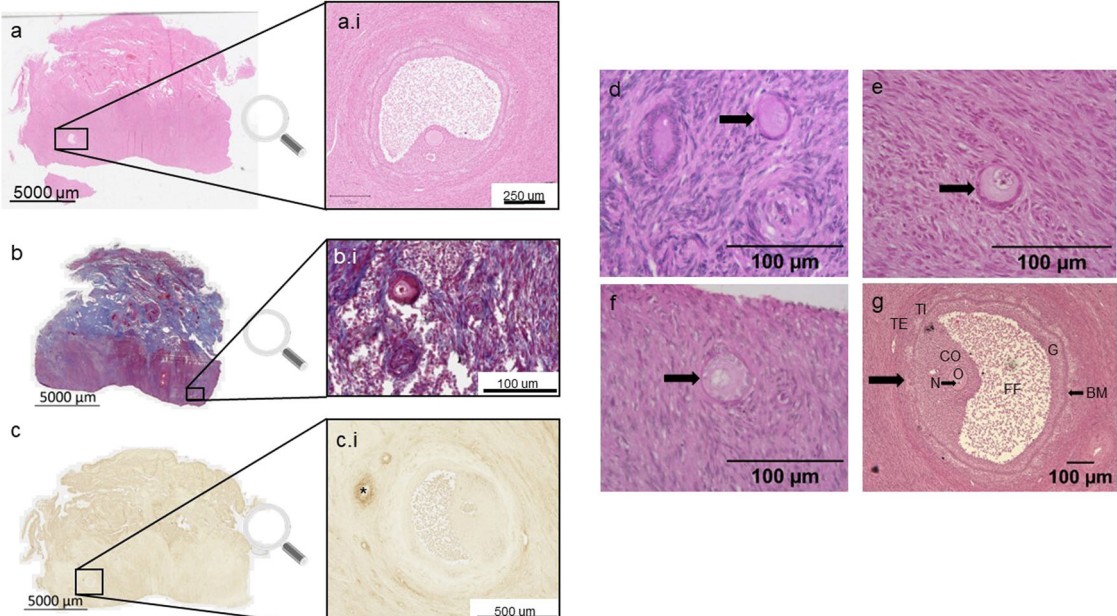

**Fig. 3 Identification of follicles in scanned ovarian sections from an adult southern white rhinoceros.** Sections have been stained with Haematoxylin and Eosin (**a**), Masson's Trichrome (**b**), and hyaluronic acid binding protein (**c**). This figure illustrates how scanned images are necessary to locate follicles in very large sections. **a**.i and **c**.i are antral follicles and **b**.i illustrates a primary follicle. **d–g** Hematoxylin and eosin staining of different follicle stages in ovarian tissue of an adult southern white rhinoceros. **d** Primordial follicle. **e** Transitional follicle. **f** Primary follicle. **g** Antral follicle. The components of the antral follicle are presented in (**g**): (O) oocyte, (N) nucleus of oocyte, (CO) cumulus oophorus, (G) granulosa cells, (BM) basal membrane, (FF) follicular fluid, (TI) theca interna, (TE) theca externa.

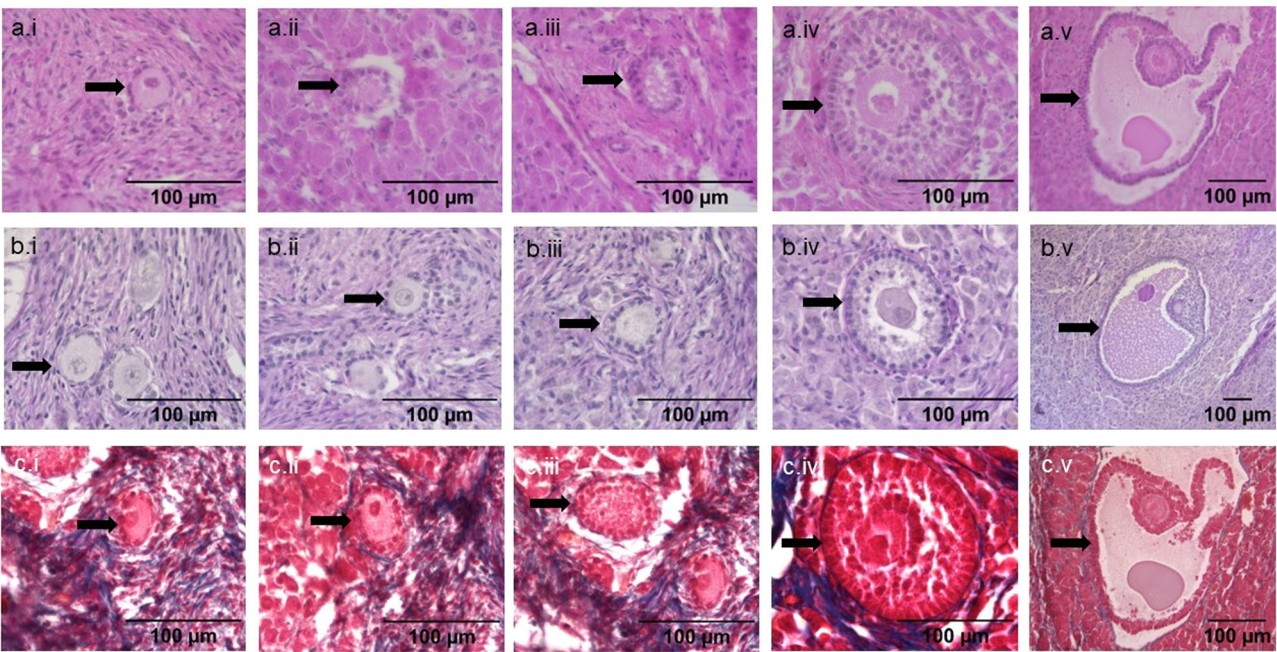

**Fig. 4 Illustration of different follicle stages in ovarian rhinoceros tissue of a neonatal southern white rhinoceros calf. a** Hematoxylin and eosin staining. **b** Periodic acid–Schiff staining. **c** Masson's trichrome staining. In (**a**), (**b**) and (**c**): i: primordial follicle, ii: primary follicle, iii: secondary follicle, iv: pre-antral follicle, v: antral follicle.

phenomenon was most apparent in the neonatal ovary because of the higher density of follicles (detailed images of the clusters in Fig. 5b.i–iv). In the adult ovary, this phenomenon was much harder to observe because of the low density of follicles. Nevertheless, in the adults, often when one follicle was found, there was a high chance of finding another one in the proximity (Fig. 3d).

**Hyaluronic acid staining**. In the neonatal ovary, hyaluronic acid (HA) was detected in follicular fluid and in the ECM around granulosa cells of smaller sized-antral follicles (Fig. 6a.i–iii). Blood vessels in the adult rhinoceros ovary were also HA rich (Fig. 6b.i). In the neonatal ovary, the basal laminae were clearly demarked by HABP staining (arrows in Fig. 6a.i–iii), whereas in the adult ovary the basal laminae of larger antral follicles were negative or very

**Table 1 Numeric parameters of different follicle stages in the neonatal and adult ovary.**

| | primordial | | transitional | | primary | | secondary | | preantral | | antral | |
|---|---|---|---|---|---|---|---|---|---|---|---|---|
| | neonatal | adult | neonatal | adult | neonatal | adult | neonatal | adult | neonatal | adult | neonatal | adult |
| Number of follicles | 9 | 1 | 12 | 8 | 3 | 4 | 7 | 4 | 1 | 0 | 2 | 2 |
| Number of granulosa cells per follicle | 9.67 ± 2.00 | 6 | 9.08 ± 4.38 | 10.50 ± 4.31 | 21.00 ± 6.93 | 16.50 ± 4.80 | 29.57 ± 13.56 | 18.25 ± 1.71 | 118 | - | 1) 252 2) 196 | 1) 521 2) 1955 |
| Mean oocyte diameter (μm) | 33.95 ± 3.28 | 30.45 | 28.82 ± 4.29 | 31.51 ± 3.28 | 32.25 ± 3.13 | 28.94 ± 3.46 | 38.65 ± 6.86 | 28.70 ± 4.03 | 50.89 | - | 1) 52.12 2) 44.56 | 1) 65.28 2) 93.08 |
| Mean follicle diameter (μm) | 43.29 ± 2.52 | 37.19 | 40.25 ± 4.59 | 43.06 ± 4.01 | 51.57 ± 8.07 | 45.41 ± 3.85 | 68.27 ± 22.34 | 46.09 ± 2.18 | 139.75 | - | 1) 338.15 2) 306.92 | 1) 310.59 2) 786.14 |
| Oocyte area (μm²) | 905.55 ± 175.76 | 704.27 | 654.80 ± 220.79 | 746.46 ± 145.55 | 853.61 ± 141.67 | 693.52 ± 157.65 | 1208.41 ± 480.33 | 621.64 ± 178.72 | 2026.05 | - | 1) 2007.15 2) 1779.11 | 1) 3383.36 2) 6798.60 |
| Follicle area (μm²) | 1479.71 ± 149.02 | 999.21 | 1229.03 ± 301.75 | 1398.49 ± 235.43 | 2115.77 ± 644.43 | 1687.61 ± 307.21 | 3690.44 ± 2166.60 | 1623.87 ± 146.38 | 14510.46 | - | 1) 78678.93 2) 76138.57 | 1) 76797.14 2) 511574.78 |
| Ratio of oocyte size to follicle size | 0.61 ± 0.09 | 0.70 | 0.53 ± 0.08 | 0.55 ± 0.13 | 0.42 ± 0.07 | 0.42 ± 0.10 | 0.36 ± 0.07 | 0.34 ± 0.10 | 0.14 | - | 1) 0.03 2) 0.02 | 1) 0.04 2) 0.01 |

The results are expressed as a mean +/− standard deviation when n is 3 or more.

weakly positive for HA (Fig. 6b.ii, iii). Although follicular fluid was positive in a smaller antral follicle (Fig. 6b.ii), in one large antral follicle from rhino 2 follicular fluid, granulosa cell ECM and basal lamina were HABP negative (Fig. 6b.iii).

Based on these samples, there is a distinct difference in the presence of HA throughout the ovary in the neonatal versus the adult ovary. In the neonatal sample, HA forms the boundaries of a 'collection' of undefined ovarian cells (Fig. 6a.i., black dashed line). In the adult ovaries, although there is a clear demarcation of HA detected between the cortex and medulla (Fig. 6b.i; dashed line), HA is distributed more uniformly in the two compartments. All negative controls show that the observed signal using HABP is specific (Fig. 6).

**Demarcation of neonatal ovarian cell nests.** Neonatal sections were subjected to molecular interrogation using immunohistochemistry (Fig. 7). Collagen I (Fig. 7a) as well as HA (Figs. 2 and 6) are part of the ECM bordering the areas of cells which make up the majority of the neonatal ovary. There is a clear demarcation of the areas (Fig. 7a, asterisk) formed by collagen I (Fig. 7a, arrow) corresponding to the collagen observed in the Masson's trichrome staining (Fig. 5).

**Characterisation of ovarian cell nests in the neonatal ovary.** Some cells situated in the neonatal structures were actively proliferating, indicated by Ki-67 and MCM2 positivity (Fig. 7b, c) while being positive for the pluripotency markers SOX2 (Fig. 7d) and Oct4 (Fig. 7e). Both pluripotency factors showed a heterogeneously positive stained pattern in the cell population (Fig. 7d, e, black arrow: strongly positive, grey arrow: weaker positive). On top, the germ cell lineage specific marker DDX4 (Fig. 7f) was convincingly positive in some cord cells clustered in nests (arrow pointing to the cells surrounded by dashed line) in contrast to stromal cells. As an internal control, the oocyte in an antral follicle also stained DDX4 positively (Fig. 7f, grey arrow), but the granulosa cells and follicular fluid were negative (Fig. 7f). At higher magnification, nests of DDX4 positive cells were clearly visible (Fig. 7g) and an overview of a large part of the section demonstrated the negative stroma (black star) in comparison to the collection of DDX4 positive cell nests grouped in the cord area (dashed line) with some DDX4 positive oocytes comprised in small follicles (grey arrow) (Fig. 7h). Anti-NaKATPase antibodies, used as a plasma membrane marker, demarked the cells of the follicular components and the undefined cells, but not the stroma (Fig. 7i). The TUNEL assay highlighted cells undergoing apoptosis, but the majority of cells were negative (Fig. 7j). CD20 revealed a very small proportion of B-lymphocytes in the cord area (Fig. 7k). A considerable number of cells in the neonatal ovary indicated the presence of AMH, a hormone produced by granulosa cells with a key role in follicle development, with AMH-positive granulosa cells and AMH-positive cells in the cell areas (Fig. 7l, m, black arrows). However, not all cells within the areas were AMH-positive indicating a heterogeneous population of molecularly distinct cells (Fig. 7m, AMH-positive: black arrow; AMH-negative: black star). Interestingly, in an adjacent section subjected to detection of SOX2 using immunohistochemistry, both the AMH-positive and -negative cells were positive for SOX2 (Fig. 7n, black arrow and black star). Blood vessels were strongly positive for CB1, a moderator of progesterone production, while the cells in the nests were moderately positive compared to a negative stromal zone (Fig. 7o).

## Discussion

To our knowledge, this study is the first to describe the structure and molecular signature of any rhinoceros ovary. We specifically

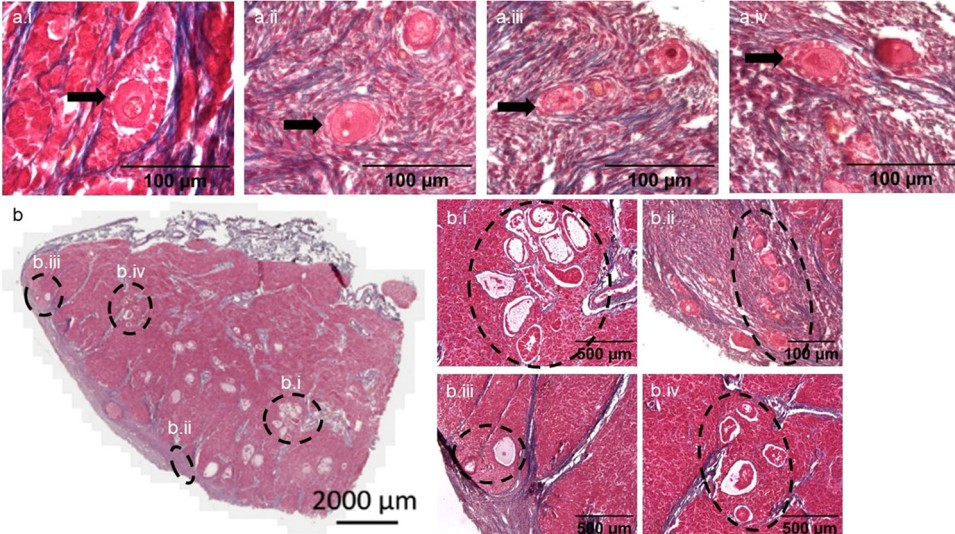

**Fig. 5 Structure and organisation of follicles in a neonatal southern white rhinoceros ovary.** Sections are stained with Masson's trichrome. (**a**.i) to (**a**.iv) Illustrations of ellipsoidal follicles in a neonatal southern white rhinoceros ovary (black arrows). **b** Presentation of follicles grouped together in a cluster in a neonatal southern white rhinoceros ovary. (**b**.i, **b**.iii, **b**.iv) Group of antral follicles (**b**.ii) Group of primary follicles.

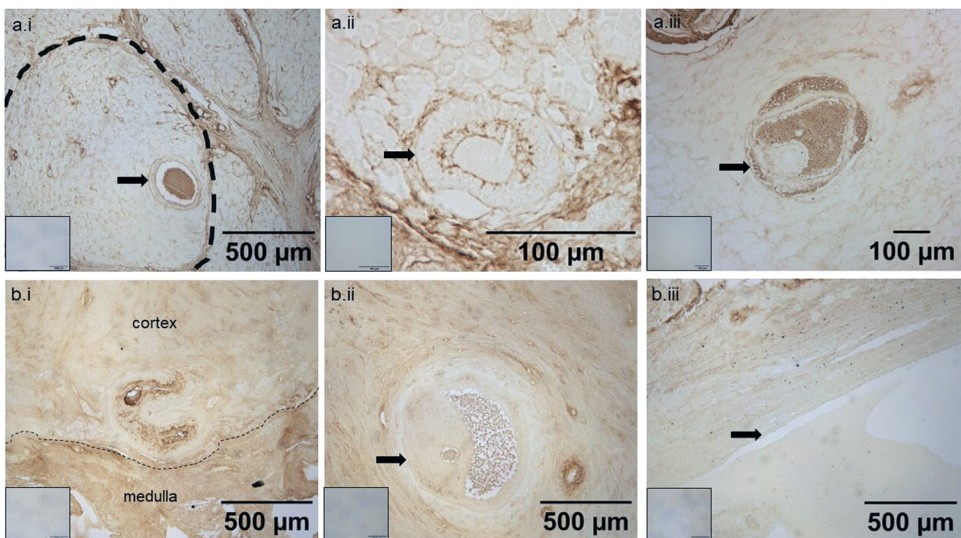

**Fig. 6 Hyaluronic acid detection in adult and neonatal southern white rhinoceros ovarian tissue. a** Neonatal calf. **b** Adult. (**a**.i) Low magnification image of distribution of hyaluronic acid in the ovary as detected using hyaluronic acid binding protein. The blacked dashed line is indicating the boundary of the collection of undefined cells which also contains a follicle (black arrow). (**a**.ii and **a**.iii) a pre-antral and antral follicle. (**b**.i) The dashed line illustrates the demarcation between the cortex and medulla part of the adult ovary. (**b**.ii and **b**.iii) two antral follicles. The black arrows indicate the basal laminae of the follicles. Inset images are negative controls.

investigated neonatal and adult southern white rhinoceros ovaries and follicles. The extraordinary finding that oogonia in germ cell nests are still observed in the neonatal rhinoceros ovary is crucial when considering conservation of this iconic species. The discovery of these germ cell precursor cells can shift the fertility preservation mind-set about rhinoceros from utilising adult reproductive tissues to using neonatal samples facilitating the development of in vitro gametogenesis and thus advance conservation efforts. Moreover, when considering the potential of ovarian tissue for conservation using aARTs, it is also important to note that ovaries from all adult animals, despite their advanced age, still contained follicles. The information we present in this paper is crucial to understanding species-specific ovarian physiology and in vitro culture methods for rhinoceros follicles[22], or even establishing in vitro gametogenesis methods starting with

oogonia. The establishment of these kind of technologies will revolutionise the field of conservation of endangered species when natural reproduction is failing or insufficient and provide new tools to fight extinction and loss of biodiversity[2].

The exciting discovery of this study is the structure of the neonatal rhinoceros ovary. While the adult ovary exhibits the cortex-transition-medulla orientation going from the medial to the lateral side, the neonatal ovary contains populations of cells grouped in cell areas separated by stroma. A similar structure was observed in the neonatal ovary of the Malayan tapir[23] in which the cells were defined as a group of luteinised stromal cells with hardly any oocytes except at the rim of the cortex. The same author described ovarian lutein activity in the neonatal ovaries of the Indian rhinoceros because of luteinisation of foetal ovaries attributed to the presence of unidentified gonadotropins during

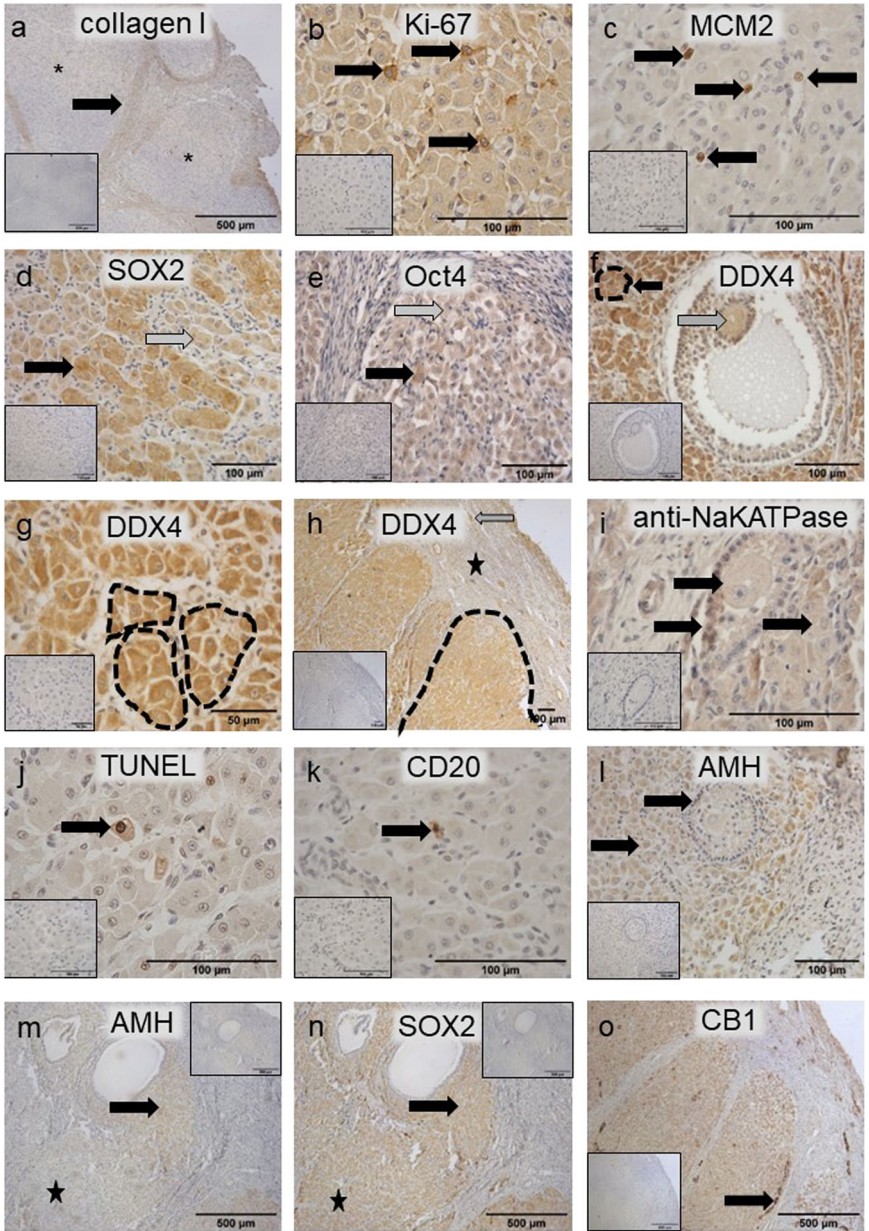

**Fig. 7 Immunohistochemical characterisation of the neonatal rhinoceros ovary.** Antibodies against several markers were applied: **a** Collagen I. **b** Ki-67. **c** MCM2. **d** SOX2. **e** Oct4. **f** DDX4. **g** NaKATPase. **h** TUNEL assay was performed. **i** CD20. **j, k** AMH. **l** SOX2. **m** CB1. **a** Collagen forms the boundaries (black arrow) between nests of cells (asterisk). **b** Ki-67 indicated some proliferation activity in the cell population (black arrow) forming the nests. **c** MCM2 indicated some proliferation activity in the cell population (black arrow) forming the nests. **d** Regions with a heterogeneity of more (black arrow) and less (grey arrow) intensely stained of SOX2-positive cells. **e** Regions with a heterogeneity of more (black arrow) and less (grey arrow) intensely stained of Oct4-positive cells. **f** An oocyte in a follicle (grey arrow) as well as the cell population building the cell nests are DDX4 positive (black arrow with dashed line surrounding the nest). **g** Dashed lines highlight DDX4 positive cell nests. **h** Oocytes at the border of the cortex are DDX4 positive (grey arrow), stroma is DDX4 negative (black star) and collections of cell nests consisting of DDX4 positive cells can be observed in the cord area (black dashed line). **i** anti-NaKATPase marked the boundaries of the cells to confirm the size (positive cell populations indicated by black arrows). **j** TUNEL positive cells were only sporadically observed in the population of cells present in the nests (black arrow). **k** Rarely a CD20 positive B cell lymphocyte was observed within the cell nests. **l** Granulosa cells as well as a large proportion of cells forming the cords are AMH positive (black arrows). **m** The nests contain a heterogeneous cell population of which some cells are AMH positive (black arrow), but other cells are AMH negative (black star). **n** For SOX2, both the AMH-positive and -negative cell populations are SOX2 positive (black arrow and black star). **o** The nests contain cells that are moderately CB1 positive. A very strong positive signal is observed in the blood vessels (black arrow). Inset images are negative controls. The scale bars in the inset images have the same length as the corresponding image.

gestation[24]. However, although the same structural aspect of large cells organised in groups by stroma can be observed in the neonatal southern white rhinoceros ovary, as we show in this study, we propose another explanation. Since the cell population is mitotically active and SOX2, Oct4 and DDX4 positive, these

structures might be ovigerous cords, also called sex cords. The pluripotent character of the cells combined with the presence of germ cell markers indicate a stage prior to fully formed oocytes. The presence of Ki-67 and MCM2 illustrates the capacity of the cells to proliferate, a feature that is not observed for oocytes, but is

for oogonia. The low rate of TUNEL-positive cells shows that there is no massive apoptosis happening yet which is expected during the meiotic prophase I stage of oocyte formation[25]. By birth, the nests of germ and somatic cells will normally break down into primordial follicles with considerable numbers of germ cells undergoing apoptosis[26] leaving individual germ cells surrounded by somatic cells. It is thus unusual that such germ cell nest structures are still observed during/after birth. It is generally accepted that folliculogenesis (or follicular histogenesis) as a last step of ovarian differentiation happens during foetal development[27]. The rat is one of the few species known for folliculogenesis to proceed after birth[28] and in the naked mole rat the presence of germ cell nests is unusually prolonged with ovaries still containing them after 3 years of age[12]. Our molecular analyses support our hypothesis that these cells in the neonatal rhinoceros ovary are germ cell nests rather than luteinized cells. Investigation of more stillborn or neonatal individuals would provide further evidence for postnatal oogenesis in the rhinoceros ovary.

The nests contain a population of cells that are AMH and SOX2 heterogeneously positive and although they look structurally the same, they have different molecular, and thus most likely functional, profiles. It is, therefore, likely that these cells either have, or are in the process of, differentiating into distinct cell populations. The meaning of the AMH positive cells observed in the cord regions of the neonatal rhinoceros ovary is unclear. In neonatal rat ovaries, AMH was found in ovarian stromal cells surrounding germ cells nests[29]. Therefore, AMH positive cells in the rhinoceros ovary around birth might point towards a type of stromal cell with a function in follicle assembly[29].

The nests of ovarian cells were also CB1 positive. In adult mice CB1 has been discovered in progesterone-producing cells of ovarian interstitial glands[30] and in the rat ovary, CB1 has been demonstrated in the ovarian surface epithelium, the granulosa cells of antral follicles and the luteal cells of functional corpora lutea[31]. Since CB1 is localised in mitochondria, where progesterone is produced, this might indicate that endocannabinoids modulate progesterone synthesis through CB1[30]. Based on these observations, we hypothesise that the CB1 positive cells in the neonatal rhinoceros ovary are steroid-producing cells.

An unusual finding revealed by this study is the flat 'two-sided' orientation of the cortex and medulla of the adult southern white rhinoceros ovary. When describing the position of the rhinoceros ovary in vivo, the hilus (defined as the site where the broad ligament contacts the ovary and is the entry point for blood vessels[32]) is located at the lateral side and the bursa is covering the medial side of the ovary[21]. This flat appearance of the southern white rhinoceros ovaries is consistent with the description of ovaries of the Sumatran rhinoceros (Dicerorhinus sumatrensis)[33]. It is likely that the age and non-reproductive condition of these animals will have affected the morphology and it is possible that ovaries of reproductive individuals containing more follicles or corpora lutea would be less flat. A 3D render mode ultrasound image of a stimulated white rhinoceros ovary showed the more typical oval-shaped ovary[34], however, as this animal was stimulated the ovary would also not have been in the natural state as it contained many more large follicles than normal. Macroscopic examination of ovaries of black and white rhinoceroses indicates that they ovulate from the cortex side of the ovary which is similar to the majority of mammals, however, this is in contrast to the rhinoceros' closest relative, the horse, which ovulate only in a specific area known as the ovulation fossa[35]. Thus, we can conclude that although phylogenetically the families of Rhinoceratidae and Equidae are relatively closely related[36], the ovarian structure is quite divergent[35,37].

In contrast to highly fecund mice, in which the ovary manifests a high follicle to stroma mass ratio, the rhinoceros ovary contains a huge amount of stroma. In the adult, HA detection defines the stroma by clearly identifying the cortex and medulla compartments with the concentration of HA higher in the medulla than the cortex. Interestingly, the granulosa cells, basal lamina and follicular fluid of small antral follicles were HA positive, but HA was not detected in granulosa cells, basal lamina or follicular fluid of larger antral follicles indicating a decline in HA function with follicle development. Similarly, in the antral follicle of the horse, the oocyte, cumulus cells, granulosa cells and theca interna were immunonegative for the HA receptor, CD44[38]. When comparing the species, in the rhinoceros sample, the theca externa was also HA negative whereas, in the mare, the theca externa was strongly immunopositive for the HA receptor[38]. The localisation of HA in the neonatal rhinoceros shows similarities with the rat embryonic ovary[39]. Both samples have the same gross distribution and are positive for HA in interstitial cells, blood vessels and ECM of granulosa cells. To strengthen our understanding of the role of HA in rhinoceros follicle development, again, more analysis with more samples will be required.

The observed low follicle density in the southern white rhinoceros is comparable to some other species such as Old World primates and ruminants[40]. When collecting samples from endangered species, samples are often from old individuals and/or pathological and thus age inherently results in a smaller follicular reserve. This reality needs to be considered when planning in vitro or in vivo culture studies. Dissection and culture methods developed for young or prime reproductive age domestic species need to be adapted because a considerable portion of the ovary may be devoid of follicles. Follicle isolation by mechanical[41] or enzymatic digestion[42] of the tissue can be considered although these protocols can damage follicles and decrease viability[14,43]. Another possibility is to develop a way to scale up the capacity of ovarian tissue culture methods to allow bulk processing of ovarian tissue. In this way, more tissue can be examined in a shorter time to allow the detection and growth of the few follicles present in the sample.

Rhinoceros follicles are constructed in the same way and consist of the same types of cells as follicles in other species. However, in the neonatal ovary, some unconventional ellipsoidal follicles were observed. In bovine follicles a similar organisation of granulosa cells has been described with most bovine primordial follicles being ellipsoidal[44]. Besides the asymmetric shape of follicles, in the neonatal ovary, follicles were often found in clusters. The observation of follicle clusters is also seen in mouse, rat, rabbit, cat, pig, tiger, marmoset[45], and human[46–48]. According to a study with human tissue, follicles within clusters develop at equal speed to solitary ones during in vitro culture and since follicles are often located in clusters, development might benefit from follicle-follicle interaction[49]. One of the proposed follicle-follicle interaction mechanisms is lateral specification which is the phenomenon that determines follicular fate by short-range cell communication[45] and this may be occurring in rhinoceros ovaries.

Many follicles in the neonatal rhinoceros ovary show signs of shrinkage and therefore abnormal morphology. Recently Adeniran et al.[50] demonstrated that shrinkage caused by fixation can lead to misinterpretation of follicle health. Because of the variability of fixation methods for these samples and the observed shrinkage artefacts, we did not perform any live/dead evaluations or measurements of the health of the follicles. Communication with zoos and collaborators to acquire samples is crucial to process the samples as quickly and with a protocol as standardised as possible[51], but force majeure in case of a sudden death of the animal or international shipping leads to deviations in such protocols.

Since we demonstrate that neonatal southern white rhinoceros ovaries contain oogonia in germ cells, the collection of neonatal ovaries with the aim of in vitro gametogenesis is a valid option. Until now, the reproductive organs of stillborn calves have not been used in fertility preservation or conservation efforts. Whereas, this study reveals that they might form a precious source of germ cells. Moreover, it is important to emphasise that ovaries from other species might also have the same potential, but not yet known. Based on this research, it is clear that investigating neonatal ovarian tissue of many more species, most crucially, endangered species, is vital to reveal their developmental state and reproductive potential. The presence of oogonia in neonatal ovaries collected post-mortem will facilitate in vitro gametogenesis technologies by circumventing the need to collect embryonic stem cells or generate induced pluripotent cells. This is, however, only the case when enough individuals are present in the population and samples of neonates can be collected. Once at the stage of critically endangered species such as the northern white rhinoceros, interventions based on aART using induced pluripotent stem cells will likely be needed.

According to the information we have, this study is the first description of the structure and molecular signature of the follicles and ovary of both adult and neonatal southern white rhinoceros. The neonatal ovary has an unusual structure that consists of cell areas containing mitotically active, pluripotent cells with germ cell characteristics, in all probability oogonia in germ cell nests. Those cells are a precious resource for in vitro gametogenesis; one of the most innovative and advanced goals in reproductive biology and regenerative medicine. Ovaries from adult rhinoceros have an unusual one-sided structure but despite advancing age, all contained follicles. Thus, by using ovarian tissue from deceased southern white rhinoceroses (neonatal and adult) to establish an in vitro culture method for germ cells or follicles, we may work towards a way to save the northern white rhinoceros – and potentially, other endangered species – from extinction.

## Methods

**Collection of ovarian tissue**. Ovarian tissue was collected from one stillborn neonatal female (rhino 1) and three adult rhinoceros (rhinos 2–4) after they were euthanized due to health reasons (Table 2). Since no infectious or congenital causes could be detected upon post-mortem investigation of the stillborn calf, the probable cause of death was determined to be perinatal hypoxia. The day after birth, its ovaries were transported to the laboratory where one was dissected into pieces and fixed (Fig. 1c). The ovaries were in a healthy state based on a macroscopic observation. In 2012, rhino 2 was treated twice with Improvac (gonadotropin-releasing-factor-analogue-protein conjugate) for contraceptive purposes. Since then, she had not shown any cycling activity. Before the treatment, she produced one calf in 2001 and one in 2005. The macroscopic view of the ovaries is illustrated in Fig. 1. Rhino 3 (Fig. 1d) did not produce any calves and no reproductive history is known about rhino 4 (Fig. 1e.i,

e.ii). After euthanasia of the adults, the ovaries were dissected, kept on ice and transported to the laboratory. After arrival, cross sections through the ovary or whole ovaries were fixed for histology.

**Histology**. Ovarian slices of all rhinoceroses (rhino 1–4; Table 2) were fixed in either Bouin's fixative, 4% paraformaldehyde, 10% (v/v) neutral buffered formalin (NBF) or Form-Acetic (NBF with 5% acetic acid)[50]; allocation of which pieces were fixed is described in Fig. 1b.

Fixed samples were embedded in paraffin wax and sectioned at 5 µm. Sections were stained with hematoxylin and eosin (H&E, Hematoxylin solution Gill No. 2, GHS232, Sigma Aldrich; eosin Y solution alcoholic with phloxine, HT110332, Sigma Aldrich), Periodic acid Schiff stain (PAS; Periodic acid solution, Sigma Aldrich, UK; 3951; Schiff's reagent, Merck, UK; 1.09033.0500) and Masson's trichrome stain (Abcam, UK; ab150686).

**Follicle parameters**. Measurements on the different follicle stages were performed in the neonatal and adult ovaries. The number of granulosa cells (Fig. 1f, yellow labelled dots), mean oocyte diameter (µm), mean follicle diameter (µm), oocyte area (µm$^2$) (Fig. 1f, area within the green dashed line), follicle area (µm$^2$) (Fig. 1f, area within the orange dotted line) and ratio of oocyte size to follicle size were recorded. All measurements were determined using ImageJ (ImageJ 1.53i, National Institutes of Health, USA) with the 'Measure' tool or 'Multi-point tool'. When three or more follicles were present in the same follicle class, the mean +/− standard deviation was calculated. Only follicles with a visible oocyte nucleus stained by H&E, PAS or Masson's trichrome were investigated. The mean oocyte diameter and mean follicle diameter were calculated by measuring the largest and perpendicular diameter (Fig. 1f, green lines for oocyte and orange lines for follicle) and subsequently taking the mean of those two measurements. When shrinkage artefact was observed, measurements were performed including the white space as illustrated in Fig. 1f (asterisk).

**Hyaluronic acid detection**. Hyaluronic acid was detected using HABP as previously carried out[52] with some modifications: 5 µg/ml HABP[53] (Calbiochem, Sigma Aldrich 385911) diluted in blocking solution for 2 h at room temperature (RT). Negative controls were blocking solution only. After washing, detection was visualised using the Vectastain ABC Elite kit (Vector laboratories, PK-6100) and a DAB peroxidase substrate kit (Vector laboratories, SK-4100).

**Immunohistochemistry**. Immunohistochemistry was performed on fixed sections for the antibody targets described in Table 3. In addition to Ki-67 we also used MCM2 to detect proliferating cells; considered to be useful for infrequently dividing cells such as granulosa cells in early pre-antral follicles[54]. Following dewaxing and rehydration, a heat-induced, low pH antigen retrieval using

**Table 2 Characteristics of the southern white rhinoceros whose ovaries were used in this study.**

| Number | Age | Date of death | Origin | Reason of death |
|---|---|---|---|---|
| rhino 1 | 0 days | January 2016 | West Midland Safari Park, United Kingdom | stillbirth |
| rhino 2 | 39.5 years | January 2018 | Whipsnade Zoo, United Kingdom | old age and quality of life decreased too much -> euthanasia |
| rhino 3 | 30 years | July 2020 | Beekse Bergen, The Netherlands | Achilles tendon ruptured during unexpected mating -> euthanasia |
| rhino 4 | 38 years | December 2020 | Salzburg Zoo, Austria | old age and health reasons -> euthanasia |

**Table 3 Description of the primary and secondary antibodies used in this study.**

| Primary antibody | Function | Catalogue | Species | Concentration | Secondary antibody | Catalogue | Dilution |
|---|---|---|---|---|---|---|---|
| collagen I[a] | extracellular matrix protein | MA1-26771 | mouse monoclonal | 25 µg/ml | goat anti-mouse IgG[d] | BA-9200 | 1:200 |
| Ki-67[a] | proliferation marker | PA5-19462 | rabbit polyclonal | 5 µg/ml | goat anti-rabbit IgG[d] | BA-1000 | 1:200 |
| minichromosome maintenance complex component 2 (MCM2)[a] | proliferation marker | PA5-79645 | rabbit polyclonal | 2.5 µg/ml | goat anti-rabbit IgG[d] | BA-1000 | 1:200 |
| sex determining region Y-box 2 (SOX2)[a] | pluripotency factor | PA1-16968 | rabbit polyclonal | 5 µg/ml | goat anti-rabbit IgG[d] | BA-1000 | 1:200 |
| octamer-binding transcription factor 4 (Oct4 or POU5F1)[b] | pluripotency factor | 11263-1-AP | rabbit polyclonal | 3.75 µg/ml | goat anti-rabbit IgG[d] | BA-1000 | 1:200 |
| DEAD-Box Helicase 4 (DDX4 or Vasa)[a] | germ cell marker | PA5-23378 | rabbit polyclonal | 5 µg/ml | goat anti-rabbit IgG[d] | BA-1000 | 1:200 |
| anti-Müllerian hormone (AMH)[a] | hormone | PA5-35851 | rabbit polyclonal | 5 µg/ml | goat anti-rabbit IgG[d] | BA-1000 | 1:200 |
| cannabinoid receptor 1 (CB1)[b] | G protein-coupled receptor | 17978-1-AP | rabbit polyclonal | 5.5 µg/ml | goat anti-rabbit IgG[d] | BA-1000 | 1:200 |
| cluster of differentiate 20 (CD20)[a] | B-lymphocyte marker | PA5-16701 | rabbit polyclonal | 0.1925 µg/ml | goat anti-rabbit IgG[d] | BA-1000 | 1:200 |
| sodium potassium ATPase (NaKATPase)[c] | membrane marker | ab58475 | rabbit polyclonal | 10 µg/ml | goat anti-rabbit IgG[d] | BA-1000 | 1:200 |

Companies: [a]Invitrogen, [b]Proteintech, [c]Abcam, [d]Vector Laboratories.

sodium citrate (pH 6.0) was applied for collagen I, Ki-67, MCM2, SOX2, DDX4, CD20 and NaKATPase. For Oct4, AMH and CB1, antigen retrieval was heat-induced high pH Vector antigen unmasking solution (Vector Laboratories, H-3301). Sections were treated with 3% hydrogen peroxide to block endogenous peroxidase activity. To prevent non-specific binding, sections were incubated in 5% normal goat serum (NGS; Vector). Sections were incubated in primary antibodies for 2 h at RT; for negative controls, the primary antibody was omitted. Samples were incubated in secondary biotinylated antibodies as described in Table 3. Detection was visualised using the Vectastain ABC Elite kit followed by DAB (as above). For all antibodies, slides were counterstained in Harris haematoxylin.

**TUNEL assay.** Detection of double-stranded DNA breaks was performed using the FragEL™ DNA fragmentation detection kit (Merck Millipore, QIA33) according to the manufacturers' instructions.

**Microscopy.** The results of H&E, PAS, Masson's trichrome and HABP staining were analysed using a light microscope (Leica DM2500). Pictures were captured by the Lumenera Infinity 5 camera with accompanying Infinity Analyse software (Teledyne Lumenera, Nepean, Canada). Scanned images were obtained by the high performance bright-field slide scanner ZEISS Axioscan 7 at magnification 20x.

**Ovarian follicle classification.** Follicles were classified according to established criteria[50,55,56] in the following groups: primordial (oocyte surrounded by a single layer of flattened pre-granulosa cells), transitional (mixed layer of flattened and cuboidal granulosa cells surrounding the oocyte), primary (single layer of cuboidal cells surrounding the oocyte), secondary (two layers of cuboidal granulosa cells surrounding the oocyte), pre-antral (many granulosa cell layers without or with interspersed fluid filled areas) and antral (many layers of granulosa cells and a large antral cavity) follicles. Since such a small number of follicles was present, all follicles were classified.

**Statistics and reproducibility.** In the adult and neonatal samples, follicles of all developmental stages were observed. However, due to the low number of follicles that could be investigated, no statistical tests comparing the neonatal or adult individuals or follicle classes could be performed on the follicular parameters measured. This is a descriptive study for 1 neonatal and 3 adult rhinoceros ovaries. All data observed in all samples were included.

**Reporting summary.** Further information on research design is available in the Nature Portfolio Reporting Summary linked to this article.

### Data availability

The data that support the findings of this study are available from the corresponding author on reasonable request.

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

## Acknowledgements

The authors would like to acknowledge Dr. Ida Parisi, lab manager of the Histology Facility at the Kennedy Institute of Rheumatology (Oxford) for help with embedding, Dr. Christoffer Lagerholm and Dr Jana Koth, the facility manager of the Wolfson imaging centre Oxford for help with slide scanning. We would also like to acknowledge the input of Prof. N. Smart of the Department of Physiology, Anatomy and Genetics for her insight into blood vessel histology and the Zoological Society of London for providing a

rhinoceros sample. This work was funded by Fondation Hoffmann to S.A.W. supporting R.A. and A.S., the Covid-19 rebuilding research momentum fund awarded to R.A. (project reference 0009963), the Oxford Medical Research Council Doctoral Training Programme (Oxford MRC-DTP) grant awarded to B.D.B. (grant number MR/N013468/1), the Petroleum Technology Development Fund (PTDF) awarded to B.V.A., the BMBF project "BioRescue" (01LC1902A) supporting T.B.H. and S.H. and the personal support of Richard McLellan to C.G.

## Author contributions

R.A. and S.A. Williams conceived and designed the study, and carried out data interpretation and analysis, and manuscript generation. R.A., B.V.A., X.W. and B.D.B. were involved in data acquisition, with R.A. generating the bulk of the data, performing the data analysis and draughting the article. R.H., S.H., T.B.H., C.G. and S.C.M. were essential for all aspects of obtaining the rhinoceros samples. All authors, including A.S. contributed to and agreed the final draught of the manuscript.

## Competing interests

The authors declare no competing interests.
