## [Peer Review File · Communications Biology]

Reviewers' comments:

Reviewer #1 (Remarks to the Author):

This is a well written and concise manuscript.

The research focus is a histological and molecular description of southern white rhinoceros ovaries obtained from n=4 females. One of the individuals is a stillborn calf.

Novel discovery of primordial germ cell nests in stillborn ovary opening a new avenue/source for these highly coveted cells in the conservation biology field.

comments:

Lines 73-82

While not specific to white rhino, preliminary in vitro studies have been published on both Sumatran and black rhinoceros.

Stoops, O'Brien, Roth Journal of Zoo and Wildlife Medicine 2011

Stoops et al., 2011 Theriogenology

Lines 106-108: similarly, follicles were found in all ovaries during gamete rescue of Sumatran and black rhino ovaries.

Stillborn black rhino ovaries similarly showed high follicle numbers (Stoops et al, 2011 black rhino paper which was focused on in vitro aspect vs histo/molecular).

Reviewer #2 (Remarks to the Author):

This manuscript provides a comprehensive histological assessment of southern white rhinoceros ovaries and provides insight into follicular development in this species of both mature females and a neonate. The information provided within may be useful for the development and optimization of assisted reproductive technologies that could play a role in conserving endangered species.

The gathered images and data are novel and useful to those studying reproductive processes in rhinoceros species. While I agree the information described in this manuscript will be useful to those developing ARTs, this concept is somewhat over-stated in the manuscript and I recommend minimizing the emphasis on this idea. If not for the abstract, I would not have been able to determine the topic of the manuscript until the last paragraph of the introduction. Based on the first few paragraphs of the introduction alone, I was surprised to find the manuscript was not focused on developing in vitro techniques. The characterization of ovarian and follicular tissue is valuable to the field in and of itself so emphasis should be placed more what is actually described than what techniques could possibly benefit from this work in the future.

The methodology is clearly described and appropriate. I appreciate that the authors did not force a statistical analysis in a situation where it is not warranted or possible.

There are a couple sentences that were difficult to follow and could perhaps be edited for clarity, specifically: Lines 53-55 "Infertility..." and line 174-175 "This study..".

The figures are wonderful additions to the limited literature describing rhinoceros reproductive histology. I did not see a justification described for staining for AMH or CB1, this information should be

included.

Overall, I think this manuscript does a good job filling a gap in the knowledge that will be useful to those in the wildlife physiology field, particularly those focused in reproduction of large mammals.

Reviewer #3 (Remarks to the Author):

This is an important investigation into the ovary of the white rhinoceros. The plight of the Northern and Southern White Rhinos and the development of new tools for reproductive assistance will be of interest to the readers of Nature Biology. The authors analyse the ovaries from 3 Southern White rhinos and present an analysis of the follicular contents. Their major finding is the low number of follicles present in the ovary and the presence of several primary follicles in the adult female rhino ovary. There are a few problems with the analysis that need to be clarified before publication regarding the presence of germ cell nests in the neonatal ovary.

Major comment:

1. The presence of germ cell nests is not convincing:

The text reads Page 6: In the neonatal sample, HA forms the boundaries of the 'nests' of 141 undefined ovarian cells (Fig. 6A.1).

Figure 6A1 is purported to carry a 'nest'. To my eye this image looks like a single follicle. The Ddx4 staining of a purported nest near the follicle in Figure 7 F is similarly not convincing.

The analysis of naked mole rats in *Reproduction*. 2021 Jan; 161(1): 89–98 has a particularly convincing example of germ cell nests in Figure 3 and Figure 4, an analysis which also used paraffin sections. I suggest that the authors carry out immunofluorescence as was done in this article to demonstrate the presence of germ cell nests.

For Figure 7: the no primary controls need to be shown in a supplementary figure. HRP staining can lead to many artefacts in the staining of neonate and adult tissues. The negative controls are needed to indicate some specificity for these antibodies.

2. The title reads 'The presence of primordial germ cells in the neonatal rhinoceros ovary'. Primordial germ cells have a separate definition for a different embryonic stage ie, before the germ cells reach the gonad.

3. There is not enough information on the histological analysis. I do not understand how the histological analysis was carried out? What thickness of sections was used? Was the whole ovary counted or were subsequent slides counted? How was the issue of counting a single follicle many times in the analysis avoided?

4. Table 1; Please change wording to number of granulosa cells per follicle in the chart. The reader might infer you are referring to overall granulosa numbers.

Rebuttal letter to Reviewers' comments:

Dear Reviewers,

We want to thank you all for your constructive comments. We tried to respond to all of them in the best possible way. We believe the manuscript has been significantly improved by your additions. As requested by the instructions of the journal, all revisions are marked in yellow in the revised manuscript and commented on in the rebuttal letter below. The 'Revised text' is illustrating the new sentences and the 'Lines in revised manuscript' is illustrating where in the manuscript you can find the revisions.

Yours sincerely,
Suzannah Williams

Reviewer #1:

Comment 1:

This is a well written and concise manuscript.

The research focus is a histological and molecular description of southern white rhinoceros ovaries obtained from n=4 females. One of the individuals is a stillborn calf.

Novel discovery of primordial germ cell nests in stillborn ovary opening a new avenue/source for these highly coveted cells in the conservation biology field.

Answer:

We would like to thank the reviewer for this favourable comment.

Comment 2:

Lines 73-82 While not specific to white rhino, preliminary in vitro studies have been published on both Sumatran and black rhinoceros.

Stoops, O'Brien, Roth Journal of Zoo and Wildlife Medicine 2011

Stoops et al., 2011 Theriogenology

Answer:

Thank you for this remark. In the last paragraph we focussed specifically on histological findings in rhinoceros species. We have now included these references of IVM and IVF in the Sumatran and black rhinoceros in a previous paragraph discussing ART.

Revised text: 'Attempts to mature and fertilise rhinoceros' oocytes have been made in the Sumatran (Dicerorhinus Sumatrensis) and black (Diceros bicornis) rhinoceros and although the low success rate, it showed gamete rescue has promise^{10,11}.'

Lines in revised manuscript: 61-63.

Comment 3:

Lines 106-108: similarly, follicles were found in all ovaries during gamete rescue of Sumatran and black rhino ovaries.

Stillborn black rhino ovaries similarly showed high follicle numbers (Stoops et al, 2011 black rhino paper which was focused on in vitro aspect vs histo/molecular).

Answer:

Thank you for pointing this out. We have included a statement on the similarity of our results to the Sumatran and black rhino publications.

Revised text: 'These results correspond to findings in the Sumatran and black rhinoceros^{10,11}.'

Line in revised manuscript: 114.

Reviewer #2:

This manuscript provides a comprehensive histological assessment of southern white rhinoceros ovaries and provides insight into follicular development in this species of both mature females and a neonate. The information provided within may be useful for the development and optimization of assisted reproductive technologies that could play a role in conserving endangered species.

Comment 1:

The gathered images and data are novel and useful to those studying reproductive processes in rhinoceros species. While I agree the information described in this manuscript will be useful to those developing ARTs, this concept is somewhat over-stated in the manuscript and I recommend minimizing the emphasis on this idea. If not for the abstract, I would not have been able to determine the topic of the manuscript until the last paragraph of the introduction. Based on the first few paragraphs of the introduction alone, I was surprised to find the manuscript was not focused on developing in vitro techniques. The characterization of ovarian and follicular tissue is valuable to the field in and of itself so emphasis should be placed more what is actually described than what techniques could possibly benefit from this work in the future.

Answer:

Thank you for your comment. On reflection, we agree and have revised the first paragraph to emphasise the focus of this study, namely histology. We have also revised the opening of the discussion.

Revised text: 'Before being able to apply innovative and challenging aART procedures, knowledge on the basic physiology of the rhinoceros species is crucial. Therefore, this study focused on providing a comprehensive histological characterisation and assessment of adult and neonatal southern white rhinoceros ovaries and their follicular development, which will lead the way to developing in vitro techniques to culture or create rhinoceros' oocytes.' and 'This study is the first to describe the structure and molecular signature of any rhinoceros. We specifically investigated neonatal and adult southern white rhinoceros ovaries and follicles.'

Lines in revised manuscript: 45-49, 186-187.

Comment 2:

The methodology is clearly described and appropriate. I appreciate that the authors did not force a statistical analysis in a situation where it is not warranted or possible.

Answer:

Thank you for this remark. We are very glad to note that you support our reasoning on the statistical analysis.

Comment 3:

There are a couple sentences that were difficult to follow and could perhaps be edited for clarity, specifically: Lines 53-55 "Infertility..." and line 174-175 "This study..".

Answer:

Thank you. We have rephrased or split the sentences to improve clarity.

Revised text: 'When a species is already extinct in the wild, like in the case of the northern white rhinoceros, infertility in ex-situ populations (kept outside of their natural habitat) might have tremendous consequences.' and 'This study is the first to describe the structure and molecular signature of any rhinoceros ovary. We specifically investigated neonatal and adult southern white rhinoceros ovaries and follicles.'

Lines in revised manuscript: 57-59.

Lines in revised manuscript: 186-187.

Comment 4:

The figures are wonderful additions to the limited literature describing rhinoceros reproductive histology. I did not see a justification described for staining for AMH or CB1, this information should be included.

Answer:

Thank you for pointing this out. We included a description of the function of these products in the manuscript to clearly state why these molecules were stained.

Revised text: 'A considerable number of cells in the neonatal ovary indicated the presence of AMH, a hormone produced by granulosa cells with a key role in follicle development, with AMH-positive granulosa cells and AMH-positive cells in the cell areas (Fig. 7L and 7M, black arrows).' and 'Blood vessels were strongly positive for CB1, a moderator of progesterone production, while the cells in the nests were moderately positive compared to a negative stromal zone (Fig. 7O).'

Lines in the revised manuscript: 176-177.

Line in the revised manuscript: 182-183.

Comment 5:

Overall, I think this manuscript does a good job filling a gap in the knowledge that will be useful to those in the wildlife physiology field, particularly those focused in reproduction of large mammals.

Answer:

We are delighted to read this comment. Thank you.

Reviewer #3 (Remarks to the Author):

This is an important investigation into the ovary of the white rhinoceros. The plight of the Northern and Southern White Rhinos and the development of new tools for reproductive assistance will be of interest to the readers of Nature Biology. The authors analyse the ovaries from 3 Southern White rhinos and present an analysis of the follicular contents. Their major finding is the low number of follicles present in the ovary and the presence of several primary follicles in the adult female rhino ovary. There are a few problems with the analysis that need to be clarified before publication regarding the presence of germ cell nests in the neonatal ovary.

Comment 1:

Major comment:

1. The presence of germ cell nests is not convincing: The text reads Page 6: In the neonatal sample, HA forms the boundaries of the 'nests' of 141 undefined ovarian cells (Fig. 6A.1). Figure 6A1 is purported to carry a 'nest'. To my eye this image looks like a single follicle. The Ddx4 staining of a purported nest near the follicle in Figure 7 F is similarly not convincing.

The analysis of naked mole rats in *Reproduction*. 2021 Jan; 161(1): 89–98 has a particular convincing example of germ cell nests in Figure 3 and Figure 4, an analysis which also used paraffin sections. I suggest that the authors carry out immunofluorescence as was done in this article to demonstrate the presence of germ cell nests.

For Figure 7: the no primary controls need to be shown in a supplementary figure. HRP staining can lead to many artefacts in the staining of neonate and adult tissues. The negative controls are needed to indicate some specificity for these antibodies.

Answer:

Thank you, we have now revised the figures and legends in response to your comments as detailed below.

In Figure 6 A.1 we added a dashed black line illustrating the hyaluronic acid boundary of the collection of undefined cells. We also added this information to the figure legend. We also replaced the word 'nest' by 'collection' to avoid confusion with germ cell nests. The black arrow is pointing to the basal

membrane of a follicle which is described in the figure legend; the legend has been revised to clarify these various aspects.

Revised text: 'In the neonatal sample, HA forms the boundaries of a 'collection' of undefined ovarian cells (Fig. 6A.1., black dashed line).' And 'The blacked dashed line is indicating the boundary of the collection of undefined cells which also contains a follicle (black arrow).'

Lines in the revised manuscript: 147-148, 495-496.

Regarding the identification of germ cell nests using DDX4. The fluorescent pictures in the naked mole rat publication using an antibody to *ddx4* for detection are indeed convincing. We have reviewed the images we presented that were also generated using an antibody to DDX4. We have compared them to those in the naked mole rat paper and appreciate that we could have illustrated our point better. We have reviewed our data and realised that we can illustrate the presence of germ cell nest more convincingly by the inclusion of additional images. Therefore, to reveal the germ cell nests, as stained by DDX4, we have included two extra images in Fig 7 at a higher and a lower magnification. In Fig 7F, the original image remains with a germ cell nest highlighted using a dashed line. In Fig 7G, a higher magnification is presented with three germ cell nests circled by a dashed line; this image is comparable to that in the naked mole rat paper. Lastly, in Figure 7H, we have a lower magnification image demonstrating the large areas or cords containing many germ cell nests. We have added an explanation of those pictures to the figure legend and also to the text.

Revised text: 'On top, the germ cell lineage specific marker DDX4 (Fig. 7F) was convincingly positive in some cord cells clustered in nests (arrow pointing to the cells surrounded by dashed line) in contrast to stromal cells. As an internal control, the oocyte in an antral follicle also stained DDX4 positively (Fig. 7F, grey arrow), but the granulosa cells and follicular fluid were negative (Fig. 7F). At higher magnification, nests of DDX4 positive cells were clearly visible (Fig. 7G) and an overview of a large part of the section demonstrated the negative stroma (black star) in comparison to the collection of DDX4 positive cell nests grouped in the cord area (dashed line) with some DDX4 positive oocytes comprised in small follicles (grey arrow) (Fig. 7H).' and '(F) An oocyte in a follicle (grey arrow) as well as the cell population building the cell nests are DDX4 positive (black arrow with dashed line surrounding the nest). (G) Dashed lines highlight DDX4 positive cell nests. (H) Oocytes at the border of the cortex are DDX4 positive (grey arrow), stroma is DDX4 negative (black star) and collections of cell nests consisting of DDX4 positive cells can be observed in the cord area (black dashed line).'

Lines in revised manuscript: 164-171, 509-513.

Regarding negative controls, (ie, 'no antibody') for HRP, we fully agree negative controls are essential for interpretation of specific staining. In Figure 7, please note that 'no primary control' images are included in the corner of all panels in the figure; as indicated in the figure legend. Having the negative control directly next to the test sample facilitates comparison as opposed to being in supplementary material. From the absence of signal in the negative control images it can be concluded in our HRP experiments no artefacts were observed and the antibodies bound specifically to the molecules investigated.

Comment 2:

2. The title reads 'The presence of primordial germ cells in the neonatal rhinoceros ovary'. Primordial germ cells have a separate definition for a different embryonic stage ie, before the germ cells reach the gonad.

Answer:

Thank you for this remark. We agree and have adapted our terminology from 'primordial germ cells' to 'oogonia in germ cell nests'. We hope this better corresponds to the used terminology in the field.

Revised manuscript: we adapted the terminology throughout the manuscript, including the title. Every replacement is marked in yellow.

Comment 3:

3. There is not enough information on the histological analysis. I do not understand how the histological analysis was carried out? What thickness of sections was used? Was the whole ovary counted or were subsequent slides counted? How was the issue of counting a single follicle many times in the analysis avoided?

Answer:

The thickness of the sections was 5 μm (added to manuscript). Counting all sections of a whole rhinoceros ovary is practically impossible. The ovary can be 10 cm long which would generate 2 000 000 sections. As illustrated in Fig. 1B and addressed on line 346, we selected an end, a middle and another end slice to be investigated by histology and immunohistochemistry. A large number of sections were stained and investigated. However, as described in the manuscript there were very small numbers of follicles visible. That is why we did not count follicles, but we described parameters of the rare follicles detected. We realise we cannot claim anything on the density of follicles, but that was not the purpose of the manuscript. We want to provide basic information on follicle appearance in the rhinoceros ovary.

To avoid counting one follicle multiple times, we only investigated follicles with a visible oocyte nucleus and these were stained by H&E, PAS or Masson's trichrome (explained in the follicle parameter section line 361-362)

Revised text: 'Fixed samples were embedded in paraffin wax and sectioned at 5 μm .'

Lines in the revised manuscript: 349.

Comment 4:

4. Table 1; Please change wording to number of granulosa cells per follicle in the chart. The reader might infer you are referring to overall granulosa numbers.

Answer:

Thank you for your comment. We adapted the wording in the table.

Revised text: 'number of granulosa cells per follicle'

Line in revised manuscript: 526 (Table 1)

REVIEWERS' COMMENTS:

Reviewer #2 (Remarks to the Author):

The authors have addressed all concerns with the first draft and I look forward to seeing this manuscript published.

Reviewer #3 (Remarks to the Author):

The authors have done a good job of revising the manuscript. I have no further comments.